# Minimax Estimation of Bandable Precision Matrices

**Addison J. Hu**[*]
Department of Statistics and Data Science
Yale University
New Haven, CT 06520
addison.hu@yale.edu

**Sahand N. Negahban**
Department of Statistics and Data Science
Yale University
New Haven, CT 06520
sahand.negahban@yale.edu

## Abstract

The inverse covariance matrix provides considerable insight for understanding statistical models in the multivariate setting. In particular, when the distribution over variables is assumed to be multivariate normal, the sparsity pattern in the inverse covariance matrix, commonly referred to as the precision matrix, corresponds to the adjacency matrix representation of the Gauss-Markov graph, which encodes conditional independence statements between variables. Minimax results under the spectral norm have previously been established for covariance matrices, both sparse and banded, and for sparse precision matrices. We establish minimax estimation bounds for estimating banded precision matrices under the spectral norm. Our results greatly improve upon the existing bounds; in particular, we find that the minimax rate for estimating banded precision matrices matches that of estimating banded covariance matrices. The key insight in our analysis is that we are able to obtain barely-noisy estimates of $k \times k$ subblocks of the precision matrix by inverting slightly wider blocks of the empirical covariance matrix along the diagonal. Our theoretical results are complemented by experiments demonstrating the sharpness of our bounds.

## 1   Introduction

Imposing structure is crucial to performing statistical estimation in the high-dimensional regime where the number of observations can be much smaller than the number of parameters. In estimating graphical models, a long line of work has focused on understanding how to impose sparsity on the underlying graph structure.

Sparse edge recovery is generally not easy for an arbitrary distribution. However, for Gaussian graphical models, it is well-known that the graphical structure is encoded in the inverse of the covariance matrix $\Sigma^{-1} = \Omega$, commonly referred to as the precision matrix [12, 14, 3]. Therefore, accurate recovery of the precision matrix is paramount to understanding the structure of the graphical model. As a consequence, a great deal of work has focused on sparse recovery of precision matrices under the multivariate normal assumption [8, 4, 5, 17, 16]. Beyond revealing the graph structure, the precision matrix also turns out to be highly useful in a variety of applications, including portfolio optimization, speech recognition, and genomics [12, 23, 18].

Although there has been a rich literature exploring the sparse precision matrix setting for Gaussian graphical models, less work has emphasized understanding the estimation of precision matrices under additional structural assumptions, with some exceptions for block structured sparsity [10] or bandability [1]. One would hope that extra structure should allow us to obtain more statistically efficient solutions. In this work, we focus on the case of bandable precision matrices, which capture

---

[*]Addison graduated from Yale in May 2017. Up-to-date contact information may be found at http://huisaddison.com/.

a sense of locality between variables. Bandable matrices arise in a number of time-series contexts and have applications in climatology, spectroscopy, fMRI analysis, and astronomy [9, 20, 15]. For example, in the time-series setting, we may assume that edges between variables $X_i, X_j$ are more likely when $i$ is temporally close to $j$, as is the case in an auto-regressive process. The precision and covariance matrices corresponding to distributions with this property are referred to as bandable, or tapering. We will discuss the details of this model in the sequel.

**Past work:** Previous work has explored the estimation of both bandable covariance and precision matrices [6, 15]. Closely related work includes the estimation of sparse precision and covariance matrices [3, 17, 4]. Asymptotically-normal entrywise precision estimates as well as minimax rates for operator norm recovery of sparse precision matrices have also been established [16]. A line of work developed concurrently to our own establishes a matching minimax lower bound [13].

When considering an estimation technique, a powerful criterion for evaluating whether the technique performs optimally in terms of convergence rate is minimaxity. Past work has established minimax rates of convergence for sparse covariance matrices, bandable covariance matrices, and sparse precision matrices [7, 6, 4, 17].

The technique for estimating bandable covariance matrices proposed in [6] is shown to achieve the optimal rate of convergence. However, no such theoretical guarantees have been shown for the bandable precision estimator proposed in recent work for estimating sparse and smooth precision matrices that arise from cosmological data [15].

Of note is the fact that the minimax rate of convergence for estimating sparse covariance matrices matches the minimax rate of convergence of estimating sparse precision matrices. In this paper, we introduce an adaptive estimator and show that it achieves the optimal rate of convergence when estimating bandable precision matrices from the banded parameter space (3). We find, satisfyingly, that analogous to the sparse case, in which the minimax rate of convergence enjoys the same rate for both precision and covariance matrices, the minimax rate of convergence for estimating bandable precision matrices matches the minimax rate of convergence for estimating bandable covariance matrices that has been established in the literature [6].

**Our contributions:** Our goal is to estimate a banded precision matrix based on $n$ i.i.d. observations. We consider a parameter space of precision matrices $\Omega$ with a power law decay structure nearly identical to the bandable covariance matrices considered for covariance matrix estimation [6]. We present a simple-to-implement algorithm for estimating the precision matrix. Furthermore, we show that the algorithm is minimax optimal with respect to the spectral norm. The upper and lower bounds given in Section 3 together imply the following optimal rate of convergence for estimating bandable precision matrices under the spectral norm. Informally, our results show the following bound for recovering a banded precision matrix with bandwidth $k$.

**Theorem 1.1** (Informal)**.** *The minimax risk for estimating the precision matrix $\Omega$ over the class $\mathcal{P}_\alpha$ given in* (3) *satisfies:*

$$\inf_{\hat\Omega} \sup_{\mathcal{P}_\alpha} \mathbf{E} \left\| \hat\Omega - \Omega \right\|^2 \approx \frac{k + \log p}{n} \tag{1}$$

*where this bound is achieved by the tapering estimator $\hat\Omega_k$ as defined in Equation* (7)*.*

An important point to note, which is shown more precisely in the sequel, is that the rate of convergence as compared to sparse precision matrix recovery is improved by a factor of $\min(k \log(p), k^2)$.

We establish a minimax upper bound by detailing an algorithm for obtaining an estimator given observations $\mathbf{x}_1, \ldots, \mathbf{x}_n$ and a pre-specified bandwidth $k$, and studying the resultant estimator's risk properties under the spectral norm. We show that an estimator using our algorithm with the optimal choice of bandwidth attains the minimax rate of convergence with high probability.

To establish the optimality of our estimation routine, we derive a minimax lower bound to show that the rate of convergence cannot be improved beyond that of our estimator. The lower bound is established by constructing subparameter spaces of (3) and applying testing arguments through Le Cam's method and Assouad's lemma [22, 6].

To supplement our analysis, we conduct numerical experiments to explore the performance of our estimator in the finite sample setting. The numerical experiments confirm that even in the finite sample case, our proposed estimator exhibits the minimax rate of convergence.

The remainder of the paper is organized as follows. In Section 2, we detail the exact model setting and introduce a blockwise inversion technique for precision matrix estimation. In Section 3, theorems establishing the minimaxity of our estimator under the spectral norm are presented. An upper bound on the estimator's risk is given in high probability with the help of a result from set packing. The minimax lower bound is derived by way of a testing argument. Both bounds are accompanied by their proofs. Finally, in Section 4, our estimator is subjected to numerical experiments. Formal proofs of the theorems may be found in the longer version of the paper [11].

**Notation:**   We will now collect notation that will be used throughout the remaining sections. Vectors will be denoted as lower-case $\mathbf{x}$ while matrices are upper-case $A$. The spectral or operator norm of a matrix is defined to be $\|A\| = \sup_{\mathbf{x}\neq 0, \mathbf{y}\neq 0}\langle A\mathbf{x}, \mathbf{y}\rangle$ while the matrix $\ell_1$ norm of a symmetric matrix $A \in \mathbf{R}^{m\times m}$ is defined to be $\|A\|_1 = \max_j \sum_{i=1}^m |A_{ij}|$.

# 2   Background and problem set-up

In this section we present details of our model and the estimation procedure. If one considers observations of the form $\mathbf{x}_1, \ldots, \mathbf{x}_n \in \mathbf{R}^p$ drawn from a distribution with precision matrix $\Omega_{p\times p}$ and zero mean, the goal then is to estimate the unknown matrix $\Omega_{p\times p}$ based on the observations $\{\mathbf{x}_i\}_{i=1}^n$. Given a random sample of $p$-variate observations $\mathbf{x}_1, \ldots, \mathbf{x}_n$ drawn from a multivariate distribution with population covariance $\Sigma = \Sigma_{p\times p}$, our procedure is based on a tapering estimator derived from blockwise estimates for estimating the precision matrix $\Omega_{p\times p} = \Sigma^{-1}$.

The maximum likelihood estimator of $\Sigma$ is

$$\hat{\Sigma} = (\hat{\sigma}_{ij})_{1\leq i,j\leq p} = \frac{1}{n}\sum_{l=1}^n (\mathbf{x}_l - \bar{\mathbf{x}})(\mathbf{x}_l - \bar{\mathbf{x}})^\top \tag{2}$$

where $\bar{\mathbf{x}}$ is the empirical mean of the vectors $\mathbf{x}_i$. We will construct estimators of the precision matrix $\Omega = \Sigma^{-1}$ by inverting blocks of $\hat{\Sigma}$ along the diagonal, and averaging over the resultant subblocks.

Throughout this paper we adhere to the convention that $\omega_{ij}$ refers to the $ij^{\text{th}}$ element in a matrix $\Omega$. Consider the parameter space $\mathcal{F}_\alpha$, with associated probability measure $\mathcal{P}_\alpha$, given by:

$$\mathcal{F}_\alpha = \mathcal{F}_\alpha(M_0, M) = \left\{ \Omega : \max_j \sum_i \{|\omega_{ij}| : |i-j| \geq k\} \leq Mk^{-\alpha} \text{ for all } k, \lambda_i(\Omega) \in [M_0^{-1}, M_0] \right\} \tag{3}$$

where $\lambda_i(\Omega)$ denotes the $i^{\text{th}}$ eigenvalue of $\Omega$, with $\lambda_i \geq \lambda_j$ for all $i \leq j$. We also constrain $\alpha > 0, M > 0, M_0 > 0$. Observe that this parameter space is nearly identical to that given in Equation (3) of [6]. We take on an additional assumption on the minimum eigenvalue of $\Omega \in \mathcal{F}_\alpha$, which is used in the technical arguments where the risk of estimating $\Omega$ under the spectral norm is bounded in terms of the error of estimating $\Sigma = \Omega^{-1}$.

Observe that the parameter space intuitively dictates that the magnitude of the entries of $\Omega$ decays in power law as we move away from the diagonal. As with the parameter space for bandable covariance matrices given in [6], we may understand $\alpha$ in (3) as a rate of decay for the precision entries $\omega_{ij}$ as they move away from the diagonal; it can also be understood in terms of the smoothness parameter in nonparametric estimation [19]. As will be discussed in Section 3, the optimal choice of $k$ depends on both $n$ and the decay rate $\alpha$.

## 2.1   Estimation procedure

We now detail the algorithm for obtaining minimax estimates for bandable $\Omega$, which is also given as pseudo-code[2] in Algorithm 1.

The algorithm is inspired by the tapering procedure introduced by Cai, Zhang, and Zhou [6] in the case of covariance matrices, with modifications in order to estimate the precision matrix. Estimating

the precision matrix introduces new difficulties as we do not have direct access to the estimates of elements of the precision matrix. For a given integer $k, 1 \leq k \leq p$, we construct a tapering estimator as follows. First, we calculate the maximum likelihood estimator for the covariance, as given in Equation (2). Then, for all integers $1 - m \leq l \leq p$ and $m \geq 1$, we define the matrices with square blocks of size at most $3m$ along the diagonal:

$$\hat{\Sigma}_{l-m}^{(3m)} = (\hat{\sigma}_{ij} \mathbf{1}\{l - m \leq i < l + 2m, l - m \leq j < l + 2m\})_{p \times p} \tag{4}$$

For each $\hat{\Sigma}_{l-m}^{(3m)}$, we replace the nonzero block with its inverse to obtain $\breve{\Omega}_{l-m}^{(3m)}$. For a given $l$, we refer to the individual entries of this intermediate matrix as follows:

$$\breve{\Omega}_{l-m}^{(3m)} = (\breve{\omega}_{ij}^{l} \mathbf{1}\{l - m \leq i < l + 2m, l - m \leq j < l + 2m\})_{p \times p} \tag{5}$$

For each $l$, we then keep only the central $m \times m$ subblock of $\breve{\Omega}_{l-m}^{(3m)}$ to obtain the blockwise estimate $\hat{\Omega}_{l}^{(m)}$:

$$\hat{\Omega}_{l}^{(m)} = (\breve{\omega}_{ij}^{l} \mathbf{1}\{l \leq i < l + m, l \leq j < l + m\})_{p \times p} \tag{6}$$

Note that this notation allows for $l < 0$ and $l + m > p$; in each case, this out-of-bounds indexing allows us to cleanly handle corner cases where the subblocks are smaller than $m \times m$.

For a given bandwidth $k$ (assume $k$ is divisible by 2), we calculate these blockwise estimates for both $m = k$ and $m = \frac{k}{2}$. Finally, we construct our estimator by averaging over the block matrices:

$$\hat{\Omega}_k = \frac{2}{k} \cdot \left( \sum_{l=1-k}^{p} \hat{\Omega}_l^{(k)} - \sum_{l=1-k/2}^{p} \hat{\Omega}_l^{(k/2)} \right) \tag{7}$$

We note that within $\frac{k}{2}$ entries of the diagonal, each entry is effectively the sum of $\frac{k}{2}$ estimates, and as we move from $\frac{k}{2}$ to $k$ from the diagonal, each entry is progressively the sum of one fewer entry.

Therefore, within $\frac{k}{2}$ of the diagonal, the entries are not tapered; and from $\frac{k}{2}$ to $k$ of the diagonal, the entries are linearly tapered to zero. The analysis of this estimator makes careful use of this tapering schedule and the fact that our estimator is constructed through the average of block matrices of size at most $k \times k$.

## 2.2 Implementation details

The naive algorithm performs $O(p + k)$ inversions of square matrices with size at most $3k$. This method can be sped up considerably through an application of the Woodbury matrix identity and the Schur complement relation [21, 2]. Doing so reduces the computational complexity of the algorithm from $O(pk^3)$ to $O(pk^2)$. We discuss the details of modified algorithm and its computational complexity below.

Suppose we have $\breve{\Omega}_{l-m}^{(3m)}$ and are interested in obtaining $\breve{\Omega}_{l-m+1}^{(3m)}$. We observe that the nonzero block of $\breve{\Omega}_{l-m+1}^{(3m)}$ corresponds to the inverse of the nonzero block of $\hat{\Sigma}_{l-m+1}^{(3m)}$, which only differs by one row and one column from $\hat{\Sigma}_{l-m}^{(3m)}$, the matrix for which the inverse of the nonzero block corresponds to $\breve{\Omega}_{l-m}^{(3m)}$, which we have already computed. We may understand the movement from $\hat{\Sigma}_{l-m}^{(3m)}, \breve{\Omega}_{l-m}^{(3m)}$ to $\hat{\Sigma}_{l-m+1}^{(3m)}$ (to which we already have direct access) and $\breve{\Omega}_{l-m+1}^{(3m)}$ as two rank-1 updates. Let us view the nonzero blocks of $\hat{\Sigma}_{l-m}^{(3m)}, \breve{\Omega}_{l-m}^{(3m)}$ as the block matrices:

$$\text{NonZero}(\hat{\Sigma}_{l-m}^{(3m)}) = \begin{bmatrix} A \in \mathbf{R}^{1 \times 1} & B \in \mathbf{R}^{1 \times (3m-1)} \\ B^\top \in \mathbf{R}^{(3m-1) \times 1} & C \in \mathbf{R}^{(3m-1) \times (3m-1)} \end{bmatrix}$$

$$\text{NonZero}(\breve{\Omega}_{l-m}^{(3m)}) = \begin{bmatrix} \tilde{A} \in \mathbf{R}^{1 \times 1} & \tilde{B} \in \mathbf{R}^{1 \times (3m-1)} \\ \tilde{B}^\top \in \mathbf{R}^{(3m-1) \times 1} & \tilde{C} \in \mathbf{R}^{(3m-1) \times (3m-1)} \end{bmatrix}$$

The Schur complement relation tells us that given $\hat{\Sigma}_{l-m}^{3m}, \breve{\Omega}_{l-m}^{(3m)}$, we may trivially compute $C^{-1}$ as follows:

$$C^{-1} = \left( \tilde{C}^{-1} + B^\top A^{-1} B \right)^{-1} = \tilde{C} - \frac{\tilde{C} B^\top B \tilde{C}}{A + B \tilde{C} B^\top} \tag{8}$$

---

**Algorithm 1** Blockwise Inversion Technique

---

**function** FITBLOCKWISE($\hat{\Sigma}$, k)

    $\hat{\Omega} \leftarrow \mathbf{0}_{p \times p}$

    **for** $l \in [1-k, p)$ **do**

        $\hat{\Omega} \leftarrow \hat{\Omega} + \text{BLOCKINVERSE}(\hat{\Sigma}, k, l)$

    **end for**

    **for** $l \in [1 - \lfloor k/2 \rfloor, p)$ **do**

        $\hat{\Omega} \leftarrow \hat{\Omega} - \text{BLOCKINVERSE}(\hat{\Sigma}, \lfloor k/2 \rfloor, l)$

    **end for**

    **return** $\hat{\Omega}$

**end function**

**function** BLOCKINVERSE($\hat{\Sigma}$, m, l)                                           ▷ Obtain $3m \times 3m$ block inverse.

    $s \leftarrow \max\{l - m, 0\}$

    $f \leftarrow \min\{p, l + 2m\}$

    $M \leftarrow \left(\hat{\Sigma}\texttt{[s:f, s:f]}\right)^{-1}$

                                          ▷ Preserve central $m \times m$ block of inverse.

    $s \leftarrow m + \min\{l - m, 0\}$

    $N \leftarrow M\texttt{[s:s+m, s:s+m]}$

                                          ▷ Restore block inverse to appropriate indices.

    $P \leftarrow \mathbf{0}_{p \times p}$

    $s \leftarrow \max\{l, 0\}$

    $f \leftarrow \min\{l + m, p\}$

    $P\texttt{[s:f, s:f]} = N$

    **return** $P$

**end function**

---

by the Woodbury matrix identity, which gives an efficient algorithm for computing the inverse of a matrix subject to a low-rank (in this case, rank-1) perturbation. This allows us to move from the inverse of a matrix in $\mathbf{R}^{3m \times 3m}$ to the inverse of a matrix in $\mathbf{R}^{(3m-1) \times (3m-1)}$ where a row and column have been removed. A nearly identical argument allows us to move from the $\mathbf{R}^{(3m-1) \times (3m-1)}$ matrix to an $\mathbf{R}^{3m \times 3m}$ matrix where a row and column have been appended, which gives us the desired block of $\breve{\Omega}_{l-m+1}^{(3m)}$.

With this modification to the algorithm, we need only compute the inverse of a square matrix of width $2m$ at the beginning of the routine; thereafter, every subsequent block inverse may be computed through simple rank one matrix updates.

### 2.3 Complexity details

We now detail the factor of $k$ improvement in computational complexity provided through the application of the Woodbury matrix identity and the Schur complement relation introduced in Section 2.2. Recall that the naive implementation of Algorithm 1 involves $O(p + k)$ inversions of square matrices of size at most $3k$, each of which cost $O(k^3)$. Therefore, the overall complexity of the naive algorithm is $O(pk^3)$, as $k < p$.

Now, consider the Woodbury-Schur-improved algorithm. The initial single inversion of a $2k \times 2k$ matrix costs $O(k^3)$. Thereafter, we perform $O(p + k)$ updates of the form given in Equation (8). These updates simply require vector matrix operations. Therefore, the update complexity on each iteration is $O(k^2)$. It follows that the overall complexity of the amended algorithm is $O(pk^2)$.

## 3 Rate optimality under the spectral norm

Here we present the results that establish the rate optimality of the above estimator under the spectral norm. For symmetric matrices $A$, the spectral norm, which corresponds to the largest singular value

of $A$, coincides with the $\ell_2$-operator norm. We establish optimality by first deriving an upper bound in high probability using the blockwise inversion estimator defined in Section 2.1. We then give a matching lower bound in expectation by carefully constructing two sets of multivariate normal distributions and then applying Assouad's lemma and Le Cam's method.

## 3.1 Upper bound under the spectral norm

In this section we derive a risk upper bound for the tapering estimator defined in (7) under the operator norm. We assume the distribution of the $\mathbf{x}_i$'s is subgaussian; that is, there exists $\rho > 0$ such that:

$$\mathbf{P}\left\{|\mathbf{v}^\top(\mathbf{x}_i - \mathbf{E}\,\mathbf{x}_i)| > t\right\} \leq e^{-\frac{t^2\rho}{2}} \tag{9}$$

for all $t > 0$ and $\|\mathbf{v}\|_2 = 1$. Let $\mathcal{P}_\alpha = \mathcal{P}_\alpha(M_0, M, \rho)$ denote the set of distributions of $\mathbf{x}_i$ that satisfy (3) and (9).

**Theorem 3.1.** *The tapering estimator $\hat{\Omega}_k$, defined in (7), of the precision matrix $\Omega_{p\times p}$ with $p > n^{\frac{1}{2\alpha+1}}$ satisfies:*

$$\sup_{\mathcal{P}_\alpha}\mathbf{P}\left\{\left\|\hat{\Omega}_k - \Omega\right\|^2 \geq C\frac{k + \log p}{n} + Ck^{-2\alpha}\right\} = O\left(p^{-15}\right) \tag{10}$$

*with $k = o(n)$, $\log p = o(n)$, and a universal constant $C > 0$.*

*In particular, the estimator $\hat{\Omega} = \hat{\Omega}_k$ with $k = n^{\frac{1}{2\alpha+1}}$ satisfies:*

$$\sup_{\mathcal{P}_\alpha}\mathbf{P}\left\{\left\|\hat{\Omega}_k - \Omega\right\|^2 \geq Cn^{-\frac{2\alpha}{2\alpha+1}} + C\frac{\log p}{n}\right\} = O\left(p^{-15}\right) \tag{11}$$

Given the result in Equation (10), it is easy to show that setting $k = n^{\frac{1}{2\alpha+1}}$ yields the optimal rate by balancing the size of the inside-taper and outside-taper terms, which gives Equation (11).

The proof of this theorem, which is given in the supplementary material, relies on the fact that when we invert a $3k \times 3k$ block, the difference between the central $k \times k$ block and the corresponding $k \times k$ block which would have been obtained by inverting the full matrix has a negligible contribution to the risk. As a result, we are able to take concentration bounds on the operator norm of subgaussian matrices, customarily used for bounding the norm of the difference of covariance matrices, and apply them instead to differences of precision matrices to obtain our result.

The key insight is that we can relate the spectral norm of a $k \times k$ subblock produced by our estimator to the spectral norm of the corresponding $k \times k$ subblock of the covariance matrix, which allows us to apply concentration bounds from classical random matrix theory. Moreover, it turns out that if we apply the tapering schedule induced by the construction of our estimator to the population parameter $\Omega \in \mathcal{F}_\alpha$, we may express the tapered population $\Omega$ as a sum of block matrices in exactly the same way that our estimator is expressed as a sum of block matrices.

In particular, the tapering schedule is presented next. Suppose a population precision matrix $\Omega \in \mathcal{F}_\alpha$. Then, we denote the tapered version of $\Omega$ by $\Omega_A$, and construct:

$$\Omega_A = (\omega_{ij} \cdot v_{ij})_{p\times p}$$
$$\Omega_B = (\omega_{ij} \cdot (1 - v_{ij}))_{p\times p}$$

where the tapering coefficients are given by:

$$v_{ij} = \begin{cases} 1 & \text{for } |i - j| < \frac{k}{2} \\ \frac{|i-j|}{k/2} & \text{for } \frac{k}{2} \leq |i - j| < k \\ 0 & \text{for } |i - j| \geq k \end{cases}$$

We then handle the risk of estimating the inside-taper $\Omega_A$ and the risk of estimating the outside-taper $\Omega_B$ separately.

Because our estimator and the population parameter are both averages over $k \times k$ block matrices along the diagonal, we may then take a union bound over the high probability bounds on the spectral norm deviation for the $k \times k$ subblocks to obtain a high probability bound on the risk of our estimator. We refer the reader to the longer version of the paper for further details [11].

## 3.2 Lower bound under the spectral norm

In Section 3.1, we established Theorem 3.1, which states that our estimator achieves the rate of convergence $n^{-\frac{2\alpha}{2\alpha+1}}$ under the spectral norm by using the optimal choice of $k = n^{\frac{1}{2\alpha+1}}$. Next we demonstrate a matching lower bound, which implies that the upper bound established in Equation (11) is tight up to constant factors.

Specifically, for the estimation of precision matrices in the parameter space given by Equation (3), the following minimax lower bound holds.

**Theorem 3.2.** *The minimax risk for estimating the precision matrix $\Omega$ over $\mathcal{P}_\alpha$ under the operator norm satisfies:*

$$\inf_{\hat{\Omega}} \sup_{\mathcal{P}_\alpha} \mathbf{E} \left\| \hat{\Omega} - \Omega \right\|^2 \geq cn^{-\frac{2\alpha}{2\alpha+1}} + c\frac{\log p}{n} \tag{12}$$

As in many information theoretic lower bounds, we first identify a subset of our parameter space that captures most of the complexity of the full space. We then establish an information theoretic limit on estimating parameters from this subspace, which yields a valid minimax lower bound over the original set.

Specifically, for our particular parameter space $\mathcal{F}_\alpha$, we identify two subparameter spaces, $\mathcal{F}_{11}, \mathcal{F}_{12}$. The first, $\mathcal{F}_{11}$, is a collection of $2^k$ matrices with varying levels of density. To this collection, we apply Assouad's lemma obtain a lower bound with rate $n^{-\frac{2\alpha}{2\alpha+1}}$. The second, $\mathcal{F}_{12}$, is a collection of diagonal matrices, to which we apply Le Cam's method to derive a lower bound with rate $\frac{\log p}{n}$.

The rate given in Theorem 3.2 is therefore a lower bound on minimax rate for estimating the union $(\mathcal{F}_{11} \cup \mathcal{F}_{12}) = \mathcal{F}_1 \subset \mathcal{F}_\alpha$. The full details of the subparameter space construction and derivation of lower bounds may be found in the full-length version of the paper [11].

# 4 Experimental results

We implemented the blockwise inversion technique in NumPy and ran simulations on synthetic datasets. Our experiments confirm that even in the finite sample case, the blockwise inversion technique achieves the theoretical rates. In the experiments, we draw observations from a multivariate normal distribution with precision parameter $\Omega \in \mathcal{F}_\alpha$, as defined in (3). Following [6], for given constants $\rho, \alpha, p$, we consider precision matrices $\Omega = (\omega_{ij})_{1 \leq i,j \leq p}$ of the form:

$$\omega_{ij} = \begin{cases} 1 & \text{for } 1 \leq i = j \leq p \\ \rho|i-j|^{-\alpha-1} & \text{for } 1 \leq i \neq j \leq p \end{cases} \tag{13}$$

Though the precision matrices considered in our experiments are Toeplitz, our estimator does not take advantage of this knowledge. We choose $\rho = 0.6$ to ensure that the matrices generated are non-negative definite.

In applying the tapering estimator as defined in (7), we choose the bandwidth to be $k = \lfloor n^{\frac{1}{2\alpha+1}} \rfloor$, which gives the optimal rate of convergence, as established in Theorem 3.1.

In our experiments, we varied $\alpha$, $n$, and $p$. For our first set of experiments, we allowed $\alpha$ to take on values in $\{0.2, 0.3, 0.4, 0.5\}$, $n$ to take values in $\{250, 500, 750, 1000\}$, and $p$ to take values in $\{100, 200, 300, 400\}$. Each setting was run for five trials, and the averages are plotted with error bars to show variability between experiments. We observe in Figure 1a that the spectral norm error increases linearly as $\log p$ increases, confirming the $\frac{\log p}{n}$ term in the rate of convergence.

Building upon the experimental results from the first set of simulations, we provide an additional sets of trials for the $\alpha = 0.2, p = 400$ case, with $n \in \{11000, 3162, 1670\}$. These sample sizes were chosen so that in Figure 1b, there is overlap between the error plots for $\alpha = 0.2$ and the other $\alpha$ regimes[3]. As with Figure 1a, Figure 1b confirms the minimax rate of convergence given in Theorem 3.1. Namely, we see that plotting the error with respect to $n^{-\frac{2\alpha}{2\alpha+1}}$ results in linear plots with almost

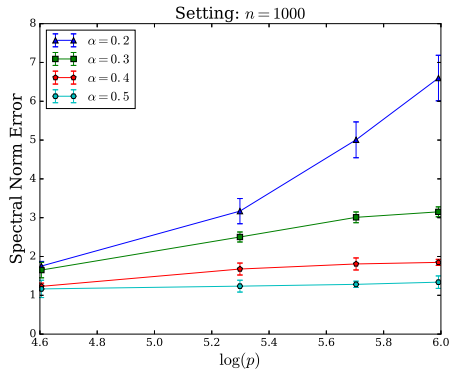

(a) Spectral norm error as $\log p$ changes.

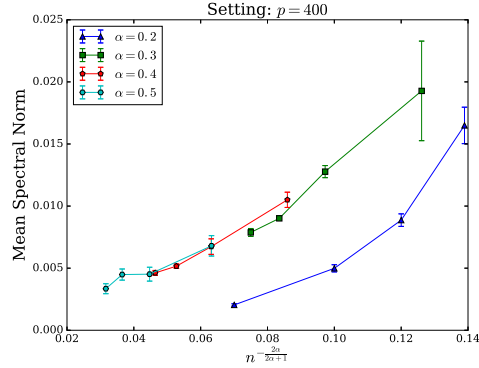

(b) Mean spectral norm error as $n^{-\frac{2\alpha}{2\alpha+1}}$ changes.

Figure 1: Experimental results. Note that the plotted error grows linearly as a function of $\log p$ and $n^{-\frac{2\alpha}{2\alpha+1}}$, respectively, matching the theoretical results; however, the linear relationship is less clear in the $\alpha = 0.2$ case, due to the subtle interplay of the error terms.

identical slopes. We note that in both plots, there is a small difference in the behavior for the case $\alpha = 0.2$. This observation can be attributed to the fact that for such a slow decay of the precision matrix bandwidth, we have a more subtle interplay between the bias and variance terms presented in the theorems above.

## 5 Discussion

In this paper we have presented minimax upper and lower bounds for estimating banded precision matrices after observing $n$ samples drawn from a $p$-dimensional subgaussian distribution. Furthermore, we have provided a computationally efficient algorithm that achieves the optimal rate of convergence for estimating a banded precision matrix under the operator norm. Theorems 3.1 and 3.2 together establish that the minimax rate of convergence for estimating precision matrices over the parameter space $\mathcal{F}_\alpha$ given in Equation (3) is $n^{-\frac{2\alpha}{2\alpha+1}} + \frac{\log p}{n}$, where $\alpha$ dictates the bandwidth of the precision matrix.

The rate achieved in this setting parallels the results established for estimating a bandable covariance matrix [6]. As in that result, we observe that different regimes dictate which term dominates in the rate of convergence. In the setting where $\log p$ is of a lower order than $n^{\frac{1}{2\alpha+1}}$, the $n^{-\frac{2\alpha}{2\alpha+1}}$ term dominates, and the rate of convergence is determined by the smoothness parameter $\alpha$. However, when $\log p$ is much larger than $n^{\frac{1}{2\alpha+1}}$, $p$ has a much greater influence on the minimax rate of convergence.

Overall, we have shown the performance gains that may be obtained through added structural constraints. An interesting line of future work will be to explore algorithms that uniformly exhibit a smooth transition between fully banded models and sparse models on the precision matrix. Such methods could adapt to the structure and allow for mixtures between banded and sparse precision matrices. Another interesting direction would be in understanding how dependencies between the $n$ observations will influence the error rate of the estimator.

Finally, the results presented here apply to the case of subgaussian random variables. Unfortunately, moving away from the Gaussian setting in general breaks the connection between precision matrices and graph structure. Hence, a fruitful line of work will be to also develop methods that can be applied to estimating the banded graphical model structure with general exponential family observations.

### Acknowledgements

We would like to thank Harry Zhou for stimulating discussions regarding matrix estimation problems. SN acknowledges funding from NSF Grant DMS 1723128.

## Footnotes

[2] In the pseudo-code, we adhere to the NumPy convention (1) that arrays are zero-indexed, (2) that slicing an array `arr` with the operation `arr[a:b]` includes the element indexed at `a` and excludes the element indexed at `b`, and (3) that if `b` is greater than the length of the array, only elements up to the terminal element are included, with no errors.

[3] For the $\alpha = 0.2, p = 400$ case, we omit the settings where $n \in \{250, 500, 750\}$ from Figure 1b to improve the clarity of the plot.

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
