[Reviews · NeurIPS 2017]

Reviewer 1



+++++++ Summary +++++++ The paper establishes the first theoretical guarantees for statistical estimation of bandable precision matrices. The authors propose an estimator for the precision matrix given a set of independent observations, and show that this estimator is minimax optimal. Interestingly the minimax rate for estimating banded precision matrices is shown to be equal to the corresponding rate for estimating banded covariance matrices. The upper bound follows by studying the behavior of inverses of small blocks along the diagonal of the covariance matrix together with classical random matrix theory results. The lower bound follows by constructing two specially defined subparameter spaces and applying testing arguments. The authors support their theoretical development via representative numerical results. +++++++ Strengths +++++++ Clarity: The paper is very nicely written and the authors appropriately convey the ideas and relevance of their work. Novelty: While I am not too familiar with the surrounding literature, the ideas in this paper appear novel. In particular, the estimator is both intuitive, as well as (relatively) easy to compute via the tricks that the authors discuss in Section 2.2. The techniques used to prove both upper and lower bounds are interesting and sound. Significance: The authors establish matching upper and lower bounds, thereby effectively resolving the question of bandable precision matrix estimation (at least in the context of Gaussian random variables). Typo: Should it be F_alpha in (3), not P_alpha? Update after authors' response: Good paper; no further comments to add.

Reviewer 2



This paper analyzes an algorithm for estimating so-called bandable precision matrices from iid samples -- vectors, drawn from a distribution with the unknown precision matrix. Error bounds are established, in terms of the spectral norm, and a complementary minimax analysis establishes the optimality of the proposed procedure. This paper is extremely well-written, concise, and complete. The actual problem being considered here fills in a "gap" in the existing literature, and is well motivated and placed well within the context of the existing literature. Overall, this should be a definite accept. In fact, this paper should serve as a model example for how to motivate, present, and discuss problems of this general flavor. I have only a few very minor points that the authors may wish to address in revision: * Section 2.2 on implementation details is welcome, given that the "naive" interpretation of the algorithm would require many matrix inversions. That said, I did find it a bit strange to discuss the implementation details in this much detail without providing some quantification of the overall implementation complexity. * The alpha=0.2 case seems to be the "outlier" in each of the experimental cases. Especially for plot 2, I might recommend omitting it to make the other three cases easier to see. In Figure 1, the growth doesn't look linear for that case, and the growth is even difficult to see in the other cases. * There seems to be mix of styles in the references, with respect to the author name convention.

Reviewer 3



The paper developes new minimax bounds on the estimation of banded precision matrices and propose a low-complexity estimation algorithm that acheive the proposed lower bound. Comments: 1. is the constraint on the minimal eigen value of \Omega necessary in the proof? since this condition is not typically met in the high-dimensional setting as n- > infty. 2. From the simulations in Fig. 1 it seems that the error in spectral norm is around 1. This can be very good or terrible depending on how large the spectral norm of the true matrix is. It would be better to write the normalized error. 3. Figure 2 is really confusing. SInce \alpha > 0 then n^{-2\alpha/(2\alpha+1)} is a decreasing function of n, which means the larger $n$ values correpond to the region very close to 0 and I do not see a constant slope in the results! May be it would be better to draw in logaritmic scale and for larger n to clearly illustrate the results. Some suggestions to improve readibility: 1. It would be better to denote vectors by boldfase small and matrices by boldface capital letters. Currently the notation is really mixed: X_i for data -- x_i 2. It is good to mention that \omega in (3) denotes the components of the matrix \Omega 3. the notation is confusing in (8) what is 1-k and 1-k/2? 4. there should be a negative sign in (9). Also it is better to unfy the notation since now in (9) we have both X and v for vectors. 5. line 64: improve the text 6. line 79: past that - beyond that